# New Therapeutic Implications of Endothelial Nitric Oxide Synthase (eNOS) Function/Dysfunction in Cardiovascular Disease

**DOI:** 10.3390/ijms20010187

**Published:** 2019-01-07

**Authors:** Andreas Daiber, Ning Xia, Sebastian Steven, Matthias Oelze, Alina Hanf, Swenja Kröller-Schön, Thomas Münzel, Huige Li

**Affiliations:** 1Center for Cardiology, Cardiology I—Laboratory of Molecular Cardiology, University Medical Center of the Johannes Gutenberg-University Mainz, 55131 Mainz, Germany; sebastiansteven@gmx.de (S.S.); matzeoelze@aol.com (M.O.); alina.hanf@gmx.net (A.H.); swenja.kroeller-schoen@gmx.de (S.K.-S.); tmuenzel@uni-mainz.de (T.M.); 2German Center for Cardiovascular Research (DZHK), Partner Site Rhine-Main, 55131 Mainz, Germany; 3Department of Pharmacology, University Medical Center of the Johannes Gutenberg-University Mainz, 55131 Mainz, Germany; xianing@uni-mainz.de

**Keywords:** cardiovascular disease, environmental stressors, life style/behavioral health risk factors, endothelial dysfunction, eNOS uncoupling, oxidative stress, inflammation

## Abstract

The Global Burden of Disease Study identified cardiovascular risk factors as leading causes of global deaths and life years lost. Endothelial dysfunction represents a pathomechanism that is associated with most of these risk factors and stressors, and represents an early (subclinical) marker/predictor of atherosclerosis. Oxidative stress is a trigger of endothelial dysfunction and it is a hall-mark of cardiovascular diseases and of the risk factors/stressors that are responsible for their initiation. Endothelial function is largely based on endothelial nitric oxide synthase (eNOS) function and activity. Likewise, oxidative stress can lead to the loss of eNOS activity or even “uncoupling” of the enzyme by adverse regulation of well-defined “redox switches” in eNOS itself or up-/down-stream signaling molecules. Of note, not only eNOS function and activity in the endothelium are essential for vascular integrity and homeostasis, but also eNOS in perivascular adipose tissue plays an important role for these processes. Accordingly, eNOS protein represents an attractive therapeutic target that, so far, was not pharmacologically exploited. With our present work, we want to provide an overview on recent advances and future therapeutic strategies that could be used to target eNOS activity and function in cardiovascular (and other) diseases, including life style changes and epigenetic modulations. We highlight the redox-regulatory mechanisms in eNOS function and up- and down-stream signaling pathways (e.g., tetrahydrobiopterin metabolism and soluble guanylyl cyclase/cGMP pathway) and their potential pharmacological exploitation.

## 1. eNOS Impairment, Vascular Dysfunction and Cardiovascular Disease: Socioeconomic Impact and Molecular Triggers

### 1.1. Global Burden of Disease Study and Cardiovascular Risk Factors

Cardiovascular diseases/events (e.g., heart failure, myocardial infarction, acute coronary syndromes) represent significant health risk factors and they are major contributors to global deaths and chronic illness/disability [1]. The Global Burden of Disease Study is a comprehensive regional and global research program of disease burden that assesses mortality and disability from major diseases, injuries, and risk factors by a collaboration of over 1800 researchers from 127 countries. Based on the data of the Global Burden of Disease Study, there was a significant shift regarding the impact of different risk factors on the global disease burden and life expectancy during the last 20 years, away from communicable childhood diseases (contagious illnesses, such as influenza, diarrhea, or meningitis) towards those for non-communicable diseases that are found in the aging population (e.g., cardiovascular diseases like heart attacks and stroke, cancers, chronic respiratory diseases, or diabetes) [2,3]. Reasonable explanations for this shift are mainly based on demographic changes and decreased childhood mortality. Socio-economic and scientific advances improved water quality, sanitation, and nutrition in most areas of the world (exceptions can be found e.g., in sub-Saharan Africa). In 2012, the Global Burden of Disease Study identified hypertension, ischemic heart disease, smoking, and cerebrovascular disease as the leading causes for premature mortality [2,3]. These risk factors and diseases are all related to cardiovascular complications and they account together for more than 50% of global deaths and more than 20% of life years lost due to premature death or spent with severe disability. A significant socioeconomic burden on societies and health care systems is the consequence of these four leading risk factors. According to the 2015 update of the Global Burden of Disease Study, these “classical” health risk factors increase the global mortality head-to-head with environmental risk factors, such as ambient particulate matter pollution and household air pollution from solid fuels, which ranked on the fifth and tenth position among all causes for global deaths [4]. Air pollution is indeed associated with a number of cardiovascular diseases [5] and environmental (traffic) noise exposure is regarded as a trigger of cerebro/cardiovascular and metabolic diseases [6], although noise-mediated cardiovascular adverse effects are less intensively characterized than those that are mediated by air pollution. Finally, mental stress (including psychosocial origins) represents another important environmental risk factor that clearly contributes to cardiovascular risk and probably also mortality, as already reported by the INTERHEART Study in 2004, by the demonstration that psychosocial stress is associated with a higher risk for myocardial infarction [7,8]. Of note, noise exposure is well-known to initiate mental stress as a first essential step in its adverse biological effects [6] and air pollution was reported to also confer central effects leading to stress-like conditions [5]. Also, mental stress itself represents a strong trigger for oxidative stress [9,10]. Most importantly, all risk factors, classical and environmental ones, ultimately lead to atherosclerosis and endothelial dysfunction, which represents an early (subclinical) correlate of atherosclerosis [11,12] (reviewed in [13]), and oxidative stress as well as inflammation represent unifying concepts for the underlying pathophysiology [14].

### 1.2. Oxidative Stress as a Unifying Molecular Trigger of Endothelial Dysfunction, Atherosclerosis, and Cardiovascular Disease

The majority of cardiovascular diseases is accompanied by an imbalance between the formation of reactive oxygen species (ROS, including superoxide, hydrogen peroxide, as well as products such as peroxynitrite or hypochlorous acid) and detoxification by low molecular weight antioxidants or ROS degrading enzymes [15,16], leading to a deviation from the steady-state [17]. The first description of the role of oxidative stress in the development and progression of cardiovascular disease in an experimental model of hypercholesterolemia was published by Harrison and Ohara [18,19]. In line with this indispensable role of oxidative stress for the development of cardiovascular disease, the uncoupling of endothelial nitric oxide synthase (eNOS) is also mediated by increased ROS formation [20] and it represents a hall-mark of most cardiovascular disease [21,22]. According to the concept of “kindling radicals” or “bonfire” hypothesis, the initial formation of superoxide (e.g., from phagocytic nicotinamide adenine dinucleotide phosphate (NADPH) oxidases, professional enzymes for synthesis of superoxide, of infiltrated leukocytes) and the subsequent formation of peroxynitrite represent the mechanism of eNOS uncoupling [23]. This kind of ROS-induced ROS formation is also well known for cross-activation of mitochondrial ROS formation by dysfunctional mitochondria [24] or the activation of xanthine oxidase and NADPH oxidases [20,25]. Regulation of eNOS activity by so-called “redox switches” is of great interest for the present review since they all may represent pharmacological targets for cardiovascular therapy: the oxidative depletion of tetrahydrobiopterin (BH_4_), oxidative disruption of the dimeric eNOS complex by oxidation of the zinc-sulfur-complex, S-glutathionylation of cysteine residues in the reductase domain, and adverse phosphorylation at Thr495/Tyr657, as well as ROS-triggered increases in levels of the endogenous eNOS inhibitor asymmetric dimethylarginine (ADMA) (for detailed review see [20,23]). The functional correlate of eNOS dysfunction/uncoupling is endothelial dysfunction in coronary and peripheral vessels, which can be measured by acetylcholine-dependent or flow-mediated dilation (FMD) and it represents an early predictor of cardiovascular events via its direct connection to the process of atherosclerosis [26,27]. Therefore, the most part of the present review will discuss (antioxidant) therapeutic interventions that prevent eNOS uncoupling, thereby normalizing endothelial function in particular and improving cardiovascular disease in general.

Oxidative stress plays a special role in the pathogenesis of arterial hypertension, which is the most important cardiovascular risk factor in industrialized countries because of its very high prevalence [2,3]. As demonstrated by Nakazono and coworkers in the early 1990’s, the administration of heparin binding superoxide dismutase significantly lowered arterial blood pressure in spontaneously hypertensive rats, suggesting a significant role of ROS for the development of hypertension [28]. Some years later, the vascular NADPH oxidase was identified as a significant superoxide source in angiotensin-II treated hypertensive mice [29] and also eNOS uncoupling was observed in this experimental model of hypertension [30]. In accordance with the above described defined “redox switches” in eNOS, increased oxidative stress within endothelial cells (e.g., induced by the NADPH oxidase [31]) causes the oxidation of BH_4_ to the ^•^BH_3_ radical and uncoupling of eNOS [32], a pathomechanism that was also observed in hypertensive mice [33]. The close connection between oxidative stress and cardiovascular prognosis is supported by a number of small cohort clinical studies (e.g., by differential effects of vitamin C infusion on FMD in patients with high or low burden of ROS formation [26]). An example is based on the significantly impaired FMD and lower levels of reduced circulating glutathione in 52 smokers versus controls [34]. Additionally, a positive correlation between FMD and superoxide dismutase activity and negative correlations between FMD and oxidized low-density lipoprotein (oxLDL)/ADMA levels in 59 patients with chronic kidney disease versus controls were reported [35]. Finally, negative correlations between vascular function (reactive hyperemia index) and malondialdehyde/8-oxo-deoxyguanosine levels in 69 patients with sleep apnea versus controls were also published [36]. Also, large clinical trials exist that support a role of oxidative stress for cardiovascular prognosis. For example, the levels of glutathione peroxidase-1 showed a positive correlation with cardiovascular event-free survival in 636 individuals [37] and the oxidative stress serum markers D-ROM (=derivatives of reactive oxygen metabolites, indicating ROS levels) and TTL (=total thiol levels, representing the redox state) were independently and strongly associated with all-cause and cardiovascular mortality in 10,622 men (Figure 1) [38]. Despite the preclinical and clinical evidence for an important role of oxidative stress for the development and progression of cardiovascular disease, there are, to date, only a few examples of clinical studies demonstrating that targeted antioxidant drugs (e.g., xanthine oxidase inhibitors, meta-analysis in 10,684 individuals) or tight control of vitamin C plasma levels (e.g., EPIC-Norfolk trial, 19,496 individuals [39]) improve the prognosis of patients with cardiovascular disease. The majority of large clinical trials failed to show any health benefit for the treatment of cardiovascular disease with non-selective antioxidant drugs [40,41], and the synthetic antioxidant drug NXY-059, despite costly development/clinical testing, failed to demonstrate any benefit in 3195 stroke patients [42].

### 1.3. Role of Inflammation for Endothelial Dysfunction and Cardiovascular Disease

Also, low-grade inflammation is associated with most cardiovascular diseases [43,44] and it contributes significantly to oxidative stress [45]. Thereby, inflammation represents an independent cardiovascular risk factor (Figure 1) [46] that can be pharmacologically targeted [47]. Therefore, we will emphasize the importance of low-grade inflammation for the development of endothelial dysfunction and cardiovascular disease as well as its close interaction with oxidative stress, not only in vascular cells, but also in perivascular adipose tissue. Most immune cells express high levels of functional NADPH oxidases and produce ROS at much higher levels than vascular cells [48,49]. This ROS formation also has importance for the function/migration of white blood cells since genetic deletion of white blood cell Nox2 (catalytic subunit of the phagocytic NADPH oxidase) prevented their infiltration to the vascular wall and the induction of vascular oxidative stress [49,50]. Activation of immune cells was also reported for mitochondrial superoxide/hydrogen peroxide formation [51,52,53]. ROS play an important role in inflammation and tissue damage [54]. By these mechanisms, oxidative stress may also contribute to the low-grade inflammation that was observed in the aging vasculature [55,56]. The contribution of B- and T-cells to the development of hypertension by angiotensin-II infusion was demonstrated by RAG-1^−/−^ mice (recombination activating gene 1 deficient mice have no mature B and T lymphocytes) [49] and hypertension/vascular oxidative stress were also attenuated by genetic ablation of myelomonocytic cells [50]. More recently, red blood cells were also reported to contribute to ROS formation in diabetes [57,58].

Systemic lupus erythematosus, rheumatoid arthritis, and also severe psoriasis represent chronic autoimmune diseases that are associated with an increased cardiovascular risk [59,60,61,62]. Psoriasis was even defined as an independent risk factor for cardiovascular disease [63] and the European League against Rheumatism guidelines even recommend cardiovascular therapy in patients with inflammatory arthritis [64]. Immune suppressive therapy of patients with rheumatoid arthritis or psoriasis led to a reduction of systolic blood pressure [65]. Patients with rheumatoid arthritis or psoriasis displayed endothelial (vascular) dysfunction [66,67]. In line with this, targeting specific cytokines or the activation of specific immune cells by anti-inflammatory therapy (e.g., monoclonal antibodies) lowered cardiovascular mortality in patients with psoriasis (interleukins IL-17/IL-23 axis) [68,69,70], systemic lupus erythematosus (IL-17A signaling) [71], and rheumatoid arthritis (cytokine IL-6, TNF-α, and IL-17A cascades) [72,73]. These data provide a mechanistic link between cardiovascular disease and the chronic autoimmune diseases (also reviewed in [59,61]).

### 1.4. Prognostic Value of Endothelial Dysfunction and Measurement in Human Subjects

Endothelial dysfunction is found in the presence of all classical cardiovascular risk factors, such as arterial hypertension [12,74], hypercholesterolemia [11], diabetes mellitus [75], obesity, and chronic smoking [76]. Endothelial dysfunction is also correlated with markers of chronic (low-grade) inflammation, such as C-reactive protein (CRP) [77] and cardiovascular risk predictors, such as adiponectin and brain natriuretic peptide (BNP) [78,79]. In addition, endothelial dysfunction was reported for the novel environmental risk factors traffic noise exposure [80,81], ambient air pollution [82,83], and mental stress [84,85]. The presence of several risk factors produces synergistic effects on endothelial function as well as the associated cardiovascular prognosis [86]. Previous studies confirmed that hypercholesterolemia or chronic smoking lead to moderate impairment of endothelial function (reduction of the maximal acetylcholine-dependent vasodilation by ~30%), whereas the presence of both risk factors caused severe endothelial dysfunction (reduction of the maximal acetylcholine-dependent vasodilation by ~ 60%) [87]. Results of the general population-based Gutenberg Health Study (5000 individuals) revealed a strong correlation between the biomarker of cardiovascular disease, pro atrial natriuretic peptide (ANP), and non-invasive measurement of conduit artery and peripheral arterial performance [88]. A meta-analysis of 15 cohort studies reported a significant correlation of endothelial function measurements using FMD with diagnosed atherosclerosis in patients [89]. These reports are contrasted by the failure of endothelial function to predict cardiovascular events in mostly healthy subjects [90] and in individuals with intermediate cardiovascular risk [91]. According to another large population-based study non-invasive measurement of vascular function failed to add prognostic value to the European Society of Cardiology risk score [92]. Therefore, measurements of intima-media thickness [90] and stiffness index [93] may be recommended for more accurate determination of endothelial and vascular function, especially since intima-media thickness also correlates with redox state and early atherosclerosis [94]. The different clinically used techniques for the measurement of the endothelial (vascular) function are summarized in Figure 2.

### 1.5. Role of eNOS in Perivascular Adipose Tissue for Vascular Function

Soltis and Cassis have reported in 1991 that the presence of perivascular adipose tissue (PVAT) reduces the contractile response of rat aorta to noradrenaline [95]. It is now a widely accepted concept that PVAT plays an important role in regulating vascular function by releasing a large number of bioactive molecules, including NO [96]. The eNOS enzyme is expressed in the PVAT, both in PVAT endothelial cells and in PVAT adipocytes (Figure 3). This has been demonstrated in immunohistochemistry analyses [97,98]. With the fluorescence imaging technique, PVAT NO production can be directly visualized in situ (Figure 4D) [98,99,100]. The amount of NO that is produced by PVAT under physiological conditions is of functional relevance and contributes significantly to total vascular NO production. The removal of PVAT reduces the basal NO production in small arteries isolated from visceral fat of healthy individuals [101]. In endothelium-denuded, PVAT-intact rat mesenteric artery, eNOS inhibition significantly enhances noradrenaline-induced contraction, indicating that PVAT-derived NO contributes to the anti-contractile effect of PVAT independently of the endothelium [102,103]. NO that is produced by the endothelial cells induces vasodilation by diffusing into the underlying smooth muscle cells and by stimulating guanylyl cyclase [104]. It is still not completely understood how PVAT-derived NO regulates vascular tone. Available data suggest that NO produced by PVAT may induce vasorelaxation through the following mechanisms [96]: by diffusing into the adjacent smooth muscle cells and activating guanylyl cyclase, by stimulating adiponectin release from PVAT adipocytes [105], or by modulating BK_Ca_ (large-conductance calcium-activated potassium) channels in smooth muscle cells and potentiating hyperpolarization [106].

Whereas, PVAT eNOS improves vascular function under physiological conditions, it may become detrimental under pathological conditions. We have recently shown that the PVAT eNOS plays even a more important role than the endothelial eNOS in obesity-induced vascular dysfunction [96]. In a mouse model of diet-induced obesity, the endothelium-dependent, NO-mediated vasodilator response to acetylcholine remains unchanged in PVAT-free aortas from mice that were fed a high-fat diet (Figure 4A–C). A vascular dysfunction of the thoracic aorta is only evident if PVAT is left intact [98,107]. Because the acetylcholine-induced vasodilation in the mouse aorta (either with or without PVAT) is completely NO-dependent, the reduced vasomotor function in the aorta of diet-induced obese mice is likely to result from eNOS dysfunction in the PVAT, but not in the endothelium. Indeed, the PVAT eNOS, but not the endothelial eNOS, was found in a dysfunctional state in obese mice [98,107]. Diet-induced obesity led to the induction of arginases in the PVAT, but not in aorta itself. The resulting L-arginine deficiency caused eNOS uncoupling in PVAT, as evidenced with eNOS-mediated superoxide production [98]. The vascular function of the aorta from obese mice could be normalized by improving L-arginine availability [98], indicating that PVAT eNOS dysfunction is causally linked to vascular dysfunction in diet-induced obesity. Moreover, eNOS phosphorylation at serine 1177 was reduced in the PVAT but not in the endothelium of diet-induced obese mice [98,107]. 

Dysfunction of PVAT eNOS can be reversed by pharmacological treatments [96,108]. In a recent study, we have shown that in vivo treatment of diet-induced obese mice with a standardized Crataegus extract completely restored the vascular function of PVAT-containing aorta [107]. For unknown reason, the effects were rather specific for PVAT [107]. The treatment improved eNOS phosphorylation in PVAT, but it had no effects on eNOS phosphorylation in the epididymal fat, nor any effects on body weight [107]. Improvement of the PVAT function has also been achieved by in vivo treatment with resveratrol [109,110] and by calorie restriction-induced sustained weight loss [96,103].

## 2. “Redox Switches” in Endothelial Nitric Oxide Synthase (eNOS) and Associated Pathways for Therapeutic Targeting

There are classical regulatory mechanisms of eNOS activity, such as calcium/calmodulin, caveolin, HSP90, palmitoylation, and myristoylation, which also control the activating phosphorylation by protein kinase B (Akt) or AMP-activated protein kinase (AMPK) at Ser1177 (Ser1179) as well as the localization of eNOS. The non-classical regulation of eNOS activity is based on the formation of redox-active species that trigger adverse phosphorylation by redox-active kinases at Thr495/Tyr657 (e.g., kinases PKC and PYK-2), S-glutathionylation, oxidative tetrahydrobiopterin depletion, dysregulation of asymmetric dimethylarginine (ADMA) formation/degradation, and disruption of the zinc-sulfur-complex stabilizing the eNOS dimer (reviewed in [20,111]). These “redox switches“ in eNOS confer alterations in enzymatic eNOS activity and they may contribute to the uncoupling of eNOS (Figure 5). eNOS uncoupling is characterized by leakage of electrons from the transport chain in the reductase domain (from NADPH over flavins FMN and FAD) and transfer to molecular oxygen yielding superoxide instead of NO, thereby switching the enzyme from a nitric oxide to a superoxide source [21,22]. NO can be antagonized by superoxide anion radicals, leading to the formation of the cytotoxic oxidant peroxynitrite, as shown by Gryglewski, Palmer and Moncada [112]. Also, down-stream of eNOS, the NO/cGMP signaling pathway contains a number of redox switches. The soluble guanylyl cyclase (sGC) can be inactivated via thiol oxidation, S-nitros(yl)ation, or ROS-triggered loss of the heme-group [13,111]. Likewise, cGMP-degrading phosphodiesterases are activated by ROS [22,113]. Other regulatory pathways of vascular function, such as endothelin-1 and prostaglandins, are under redox control. The endothelin-1 mRNA promoter is stabilized by ROS and vice versa endothelin-1 can stimulate NADPH oxidase activity and expression levels [111]. Finally, prostaglandin formation is redox-regulated by the “peroxide tone” needed for the activity of cyclooxygenase-1/2, inactivation of prostacyclin synthase by tyrosine nitration, and as a consequence of these two oxidative processes a switch of the prostanoid synthesis to thromboxane inducing a vasoconstrictory and aggregatory phenotype [111].

### 2.1. Oxidative Depletion of Tetrahydrobiopterin

Oxidative loss of tetrahydrobiopterin (BH_4_) as a trigger for eNOS uncoupling is the best characterized “redox switch” in eNOS and meanwhile well documented in hypertension, diabetes, and atherosclerosis [33,114,115,116,117,118]. Reviews on the role of BH_4_ deficiency in almost all cardiovascular diseases provide detailed insights on the mechanisms [23,119,120,121,122]. The enzymatic source for BH_4_ synthesis, GTP-cyclohydrolase-1 (GCH-1), was identified as an important regulator of eNOS and endothelial function [123]. The stoichiometry between eNOS and GCH-1 expression controls endothelial function and eNOS overexpression without matched increase in BH_4_ levels will ultimately result in eNOS uncoupling [117]. In accordance with the concept of direct oxidative depletion of BH_4_ by peroxynitrite, an oxidant-driven proteasomal degradation of the GCH-1 has been demonstrated via peroxynitrite- or superoxide-mediated activation of the proteasome 26S [124,125,126]. Recently, Chuaiphichai et al. have demonstrated that genetic endothelial-specific GCH-1 deficiency in Gch1^fl/fl^Tie2cre mice causes eNOS uncoupling (evident by increased endothelial superoxide formation) and an impaired endothelium-dependent relaxation in arterial resistance arteries [127,128]. These data indicate that endothelial GCH-1 is the essential regulator of eNOS functionality and it can prevent uncoupling of eNOS. In a just published work, the same group show that BH_4_ deficiency in endothelial and macrophages is required to trigger endothelial dysfunction and the enhancement of atherosclerosis by using endothelial/myeloid-specific GCH-1 and ApoE global knockout (Gch1^fl/fl^Tie2CreApoE^−/−^) mice and bone marrow chimeras thereof [129,130]. These data indicate that NOS isoforms in the vasculature (eNOS) and immune cells (iNOS) need tight regulation by BH_4_ in order to prevent the progression of atherosclerosis, as already suggested [131], thereby providing another strong link between vascular function, cardiovascular health, and inflammation. Besides the GCH-1 dependent de novo synthesis of BH_4_, the so called “salvage pathway”, is of high physiological importance, consisting of the recycling of oxidized BH_2_ back to BH_4_ by dihydrofolate reductase (DHFR) [132,133]. Also, DHFR is subject to 26S proteasome-dependent degradation, a process that was prevented by S-nitros(yl)ation of DHFR by eNOS-derived NO [134]. Therefore, the BH_4_ regulatory system itself provides multiple pharmacological targets for therapeutic prevention of endothelial dysfunction and the progression of cardiovascular disease (Figure 6) [135]. As suggested previously, a combination of antioxidant therapy and BH_4_ supplementation may be required in order to successfully treat cardiovascular diseases [40].

### 2.2. Oxidative Disruption of the Zinc-Sulfur-Complex (ZnCys_4_) in the Binding Region of the eNOS Dimer

Another direct redox-regulatory pathway for eNOS function is the oxidative disruption of the zinc-sulfur-complex (ZnCys_4_) in the binding region of the eNOS dimer, resulting in a loss of sodium dodecyl sulfate (SDS)-resistant eNOS dimers, which has been first described by Zou and coworkers for peroxynitrite-mediated oxidation of eNOS [136]. The reports on this “redox switch” reflected by a decreased eNOS dimer/monomer ratio were previously summarized until the year 2010 [20,23]. More recent studies reported impaired endothelial function and a decrease in eNOS dimer/monomer ratio by CRP treatment of rats [137], in old rats with improvement by arginase inhibition [138], in mild hyperhomocysteinemia in mice with heterozygous gene deletion of methylenetetrahydrofolate reductase (Mthfr^+/−^) [139], and in 8-nitro-cGMP treated mice [140]. Similar observations were made in caveolin-1 depleted and angiotensin-II stimulated endothelial cells [141], in diabetic rats with improvement by nicorandil [142], in diabetic rats with improvement by green tea extract [143], in hypoxic pigs with improvement by L-citrulline [144], and in tachycardia/sympathetic over-activation in mice by isoproterenol [145]. eNOS dimer/monomer ratio was also decreased in diabetic db/db mice and normalized by saxagliptin [146], in erectile tissue of type 2 diabetes mellitus (T2DM) men [147], in mice with doxorubicin-induced cardiomyopathy and improvement by folic acid [148], in spontaneously hypertensive rats [149], and the improvement of eNOS coupling by increased eNOS dimer/monomer ratio by exercise training [150,151]. The critical role of this zinc-sulfur-complex for proper eNOS dimer formation was demonstrated by significant monomerization in a knock-in mouse expressing a C101A-eNOS mutant with impaired zinc-sulfur-complex forming ability [152]. Of note, these C101A-eNOS-transgenic mice displayed normal systolic blood pressure despite higher levels of eNOS, whereas mice overexpressing wild type eNOS showed significant hypotension [153]. According to another hypothesis, the dimer formation is mainly regulated by BH_4_ binding in a pocket of the dimer interface and prevention of the BH_4_ binding by mutation of the critical tryptophan 447 caused monomerization of the eNOS enzyme [154]. We think that both monomerization mechanisms could play a synergistic role and oxidative disruption of zinc-sulfur-complex is a very prominent biological redox reaction, as seen for zinc-finger transcription factors, phorbol ester domain in PKC [111], or the active site of alcohol dehydrogenase [155]. More recent reports also suggest that the dimer/monomer ratio can be also used as a surrogate parameter of neuronal NOS (nNOS) uncoupling [156]. Although eNOS dimer/monomer ratio might not represent a drug-targetable pathomechanism, it might represent a surrogate marker to monitor the successful recoupling of eNOS by pharmacological or non-pharmacological therapy.

### 2.3. S-glutathionylation of the eNOS Reductase Domain

S-glutathionylation represents another important “redox switch” in eNOS. Zweier and coworkers showed that eNOS is adversely regulated and uncoupled (leading to superoxide formation) by S-glutathionylation at cysteine residues Cys689 and Cys908 in the reductase domain [157,158]. The reports on this “redox switch” that were reflected by eNOS S-glutathionylation were previously summarized until the year 2011 [20,23,43]. More recent in vivo studies reported on an association of eNOS S-glutathionylation with eNOS uncoupling and/or endothelial dysfunction in isosorbide-5-mononitrate treated mice [159], in hypertensive mice with improvement by inhibition of mitochondrial permeability transition pore opening [160], in aged mice with potentiation by glutathione peroxidase-1 deficiency [161], in atherosclerotic mice treated with carbamylated low-density lipoprotein [162], in mice with lung injury after lipopolysaccharide challenges [163], in mice with cardiac pressure-overload and improvement by N-acetylcysteine [164], in rats with T2DM and erectile dysfunction [165], and in mice with doxorubicin-induced cardiomyopathy and improvement by folic acid [148]. Also, environmental risk factors, such as traffic noise exposure, were associated with eNOS S-glutathionylation in mice exposed to aircraft noise for several days [166,167]. eNOS S-glutathionylation in mice with pressure overload produced by transverse aortic constriction was normalized by physical exercise [168]. eNOS S-glutathionylation was also observed in a knock-in mouse expressing a C101A-eNOS mutant with impaired zinc-sulfur-complex forming ability, which was also associated with increased superoxide production and protein tyrosine nitration [153]. Also, ex vivo experiments yielded similar results and eNOS S-glutathionylation with eNOS uncoupling was observed in human aortic endothelial cells that were treated with ultrafine particles [169] and in endothelial cells upon hypoxia/reoxygenation (associated with BH_4_ depletion) [170]. This redox-regulatory mechanism gained even more biological relevance when Zweier and colleagues showed that eNOS S-glutathionylation can be reversed by glutaredoxin-1 [171]. An essential role of glutaredoxin-1 for the control of eNOS S-glutathionylation was also demonstrated in mice with necrotizing enterocolitis [172,173]. eNOS S-glutathionylation in experimental fibrosis is directly connected to glutathione biosynthesis [174]. eNOS S-glutathionylation is obviously connected to BH_4_ deficiency [175] and also a direct connection between nNOS function and eNOS S-glutathionylation was shown in the heart [176]. It remains to be established whether eNOS S-glutathionylation represents a drug-targetable pathomechanism. However, it at least represents a surrogate marker to monitor the successful recoupling of eNOS by pharmacological or non-pharmacological therapy.

### 2.4. Phosphorylation at Thr495 and Tyr657

Regulation of eNOS is also mediated by phosphorylation via redox-sensitive kinases. Whereas, eNOS phosphorylation at Ser1177 via Akt pathway is of activating nature [177], phosphorylation at Tyr657 mediated by the protein tyrosine kinase-2 (PYK-2) is of inactivating nature [178]. Also, phosphorylation at Thr495 that is mediated by protein kinase C (PKC) leads to the inactivation and potential uncoupling of eNOS [179,180,181], and PKC is also a potent activator of NADPH oxidases NOX-1 and NOX-2. PKC has redox-sensitive thiols in its phorbol ester/diacylglycerol binding site and it can be activated by hydrogen peroxide [182,183]. PYK-2 was reported to be stimulated by authentic hydrogen peroxide as well as by angiotensin-II stimulated ROS formation in cell culture and in vivo [178]. The reports on this “redox switch” reflected by adverse phosphorylation by redox-sensitive kinases were previously reviewed in detail [20,23,43], but some old reports and new studies were not considered and they are highlighted here. Homocysteine obviously causes dysregulation of eNOS via PKC activation and adverse (de)phosphorylation of eNOS in human platelets [184]. Treatment of human aortic endothelial cells with CRP caused the dephosphorylation of eNOS at Ser1177 and phosphorylation at Thr495 [185]. In isolated, pre-contracted, endothelium-intact porcine coronary arteries, amlodipine increased eNOS phosphorylation at Ser1177 and it decreased the one at Thr495, which was associated with an improvement of bradykinin-dependent relaxation [186]. Likewise, the AT1-receptor blocker telmisartan improved the ratio of phosphoThr495/phosphoSer1177 of eNOS in cultured endothelial cells [187] and nitroglycerin-treated rats [188]. Simvastatin was reported to increase NO formation, eNOS phosphorylation at Ser1177, to decrease phosphorylation at Thr495, and improve cardiac function in patients undergoing cardiac surgery [189]. Other therapeutic strategies for improvement of the ratio of phosphoThr495/phosphoSer1177 in eNOS were reported for exercise of mice [190,191], inhibition of the ROCK and ERK kinase pathways by MEKK1/2 inhibitor U0126 and ROCK inhibitor Y27632, thereby preventing the oxidative activation of PKC in human umbilical vein endothelial cells (HUVECs) [192]. Conversely, there seems to be a feedback mechanism to prevent over-activation of the “NO pathway”, as demonstrated by a decrease in Ser1177 and an increase in Thr495 phosphorylation of eNOS by chronic administration of high dose inorganic nitrate, which also resulted in impaired endothelium-dependent relaxation [193], providing a possible explanation as to why chronic NO donor therapy is not suitable for targeting the NO/cGMP pathway.

### 2.5. ADMA Formation and Degradation by DDAH

Asymmetric dimethylarginine (ADMA) is a potent endogenous inhibitor of eNOS [194], which may trigger the uncoupling of eNOS [195]. ADMA serum/plasma levels have prognostic value for future cardiovascular events in patients at increased risk [196,197]. ADMA as an eNOS “redox switch” was previously reviewed in more detail and is based on increased ADMA generation (e.g., by the activation of S-adenosylmethionine-dependent protein arginine methyltransferase (PRMT, type I)) and on decreased ADMA degradation (e.g., by inhibition of hydrolyzing dimethylarginine dimethylaminohydrolase (DDAH) enzymes) under oxidative stress conditions [20,23,43]. The relationship between ADMA, eNOS activity, and oxidative stress was also highlighted by Sydow and Münzel [195]. According to an overview on circulating biomarkers in diabetic patients, ADMA may be used as an indicator of endothelial dysfunction in diabetes [198]. Rosuvastatin attenuated isoproterenol-induced hypertrophy, remodeling/dysfunction of the ventricle, and increased myocardial NO in rats, which was associated with normalized ADMA and PRMT1/DDAH2 expression levels [199]. Simvastatin increased DDAH1 expression levels and decreased ADMA concentrations in cultured endothelial cells [200]. Another study reported that coronary artery disease patients with too high ADMA serum concentrations are not protected anymore by simvastatin therapy against cardiovascular events, which however could be prevented by DDAH2 overexpression in a related experimental animal model [200]. Serum ADMA levels were significantly increased and endothelial function and DDAH activity were inhibited in hyperlipidemic rabbits, all of which were improved by therapy with the angiotensin converting enzyme (ACE) inhibitor captopril [201].

### 2.6. L-Arginine Deficiency

L-Arginine is the endogenous substrate of eNOS and “L-arginine deficiency” may contribute to the uncoupling of eNOS [202]. However, the K_m_ (the concentration of L-arginine that is necessary for half maximal saturation) of eNOS for L-arginine is approximately 2.9 µM, whereas intracellular L-arginine concentrations are usually in the mM range [20,23,43]. Therefore, L-arginine depletion as a regulator of eNOS was considered to be unlikely. Nevertheless, a number of pre-clinical and clinical studies reported on highly beneficial effects by oral L-arginine supplementation and also increased superoxide formation by isolated NOS enzymes in the absence of L-arginine was observed. The direct antioxidant effects of the guanidino-group L-arginine may be one explanation for these observations [203]. Improved export of ADMA from endothelial cells by high dose L-arginine represents another explanation, which would become even more attractive when the y^+^L amino acid transporters (y^+^LAT-1 and -2) are impaired (e.g., by genetic deficiency) and the cationic amino acid transporter (CAT-1) takes over using cationic amino acids, such as L-arginine for the export of ADMA [204]. We had described this concept in a case report [204]. Moreover, the uptake of ADMA into brain may make use of these transport mechanisms and L-arginine may beneficially modulate this uptake [205]. There is clinical evidence that pharmacological treatment of patients with acute congestive heart failure can change the expression levels of the cationic amino acid transporter (CAT-1) and thereby modulate ADMA levels in serum and tissues [206].

L-arginine deficiency can also occur as a result of arginase induction. Arginases are major L-arginine-consuming enzymes that metabolize L-arginine to urea and L-ornithine. The up-regulation of arginase limits L-arginine bioavailability for eNOS, which represents a mechanism of eNOS uncoupling [207]. In the PVAT of diet-induced obese mice, arginase expression was enhanced and L-arginine content was reduced. These changes were associated with an uncoupling of PVAT eNOS [98]. Importantly, the incubation of PVAT-containing aorta with a combination of L-arginine and an arginase inhibitor for 30 min ex vivo in organ chambers restored the vasodilator effects of PVAT-containing aortas from obese mice [98]. These results demonstrate that L-arginine deficiency is indeed a reason for PVAT eNOS dysfunction and that eNOS uncoupling is causally involved in obesity-induced vascular dysfunction [96].

## 3. Other State-of-the-Art and Future Therapeutic Strategies for Targeting eNOS Dysfunction/Uncoupling

It would be beyond the scope of this review to discuss all previous strategies for targeting endothelial dysfunction. We here provide a brief summary on general cardiovascular therapies that also improve oxidative stress, inflammation, and endothelial dysfunction, but will mainly concentrate on new or future therapeutic strategies to target eNOS dysfunction and uncoupling as well as up- and down-stream signaling molecules.

### 3.1. Established Cardiovascular Drugs: Statins, ACE-Inhibitors and AT1-Receptor Blockers 

According to a number of reports in the literature, endothelial dysfunction, atherosclerosis, and the late cardiovascular complications of these adverse phenomena are associated with a chronic activation of the local and/or circulating renin-angiotensin-aldosterone system (RAAS). Suggesting a role of an inappropriate ROS production in this setting, diabetic patients show particularly beneficial responses to mineralocorticoid receptor blockade [208], ACE inhibitor [209], AT_1_-receptor blocker [210], and renin inhibitor [211] therapy. Also, statin therapy was shown to beneficially affect diabetic cardiovascular complications [212]. These established cardiovascular drugs improve cardiovascular prognosis by lowering oxidative stress levels, inflammation, and mediators of vascular dysregulation, such as lipids and vasoconstrictors (reviewed in [1,213,214,215]). In addition, RAAS targeting drugs and statins improve NO/cGMP signaling directly by beneficially modulating the above described “redox switches” in this pathway. The beneficial pleiotropic effects of statins may be also related to the induction of the Nrf2-heme oxygenase-1 system as well as improved function of endothelial progenitor cells [216].

There are also a number of candidates of other drug classes that display highly beneficial “pleiotropic” effects. Examples are the third generation beta-blocker nebivolol that stimulated endothelial NO formation [217], thereby reverses endothelial dysfunction in patients with essential hypertension [218], improves oxidative stress in a rat model of hypertension [219], and prevents multiple complications in experimental myocardial infarction [220]. Hydralazine is one of the first antihypertensive drugs for clinical use, now mainly used for treatment of pre-eclampsia [221], and experienced a revival when a drastic improvement of heart failure mortality was reported for the combination therapy of hydralazine and isosorbide dinitrate (A-HeFT) [222,223]. Pentaerithrityl tetranitrate (PETN) is the only organic nitrate in clinical use that is devoid of the induction of side effects, such as oxidative stress, nitrate tolerance, and endothelial dysfunction [113,224]. The molecular explanation for the beneficial effects of PETN, not shared by other nitrates, is the induction of heme oxygenase-1 [225,226] as well as hundreds of other genes, in a Nrf2-dependent fashion [227]. The pleiotropic effects of these three drugs and others were discussed in detail previously [215].

### 3.2. Therapeutic Targeting of Cascades up- and Down-Stream of the eNOS/NO/sGC/cGMP Axis

Based on the above described appreciable redox-regulation of the eNOS/NO/sGC/cGMP axis in many cardiovascular diseases, this signaling pathway represents an attractive target for pharmacological therapy of various cardiovascular diseases [13]. Several of the therapeutic agents that target this signaling pathway represent real success stories, as demonstrated by the following examples. The NO replacement therapy by organic nitrates is a mainstay in cardiovascular therapy of stable angina, congestive heart failure and acute coronary syndromes, despite limitations that are due to the development of nitrate tolerance and endothelial dysfunction under chronic therapy that are mainly related to the induction of oxidative stress by most organic nitrates [113]. According to preliminary human data, some organic nitrates could be also used for the therapy of pulmonary arterial hypertension and preeclampsia. NO donors, such as sodium nitroprusside, are also used for the acute therapy of hypertensive crisis in the emergency room. However, reasons for the failure to use NO donors for chronic therapy have been already discussed above. Here, we will only briefly mention some of the mainstay drugs targeting the NO/cGMP signaling pathway and refer to more detailed information in a previous review [13]. Phosphodiesterase inhibitors prevent the enzymatic degradation of cGMP and they are used in multiple indications comprising erectile dysfunction, pulmonary hypertension, and a number of cardiac diseases and they were reported to prevent endothelial dysfunction, oxidative stress and inflammation [228]. Activators and stimulators of the sGC represent a novel class of “repair” drugs of a (oxidatively) damaged sGC enzyme and they are intended to be used for the therapy of pulmonary hypertension and heart failure, but also in other indications, such as arterial hypertension, renal fibrosis/failure, liver cirrhosis, erectile dysfunction, atherosclerosis, restenosis, thrombosis, and inflammation [229].

### 3.3. Therapeutic Targeting of eNOS Enzyme

The improvement of endothelial function (ex vivo) by sepiapterin [118] suggests that this drug could be also used to prevent eNOS uncoupling in vascular tissue in vivo. An example is provided in Figure 7, displaying the highly beneficial effect of ex vivo incubation with sepiapterin/PEG-SOD on vascular reactivity of aortic ring segments from angiotensin-II infused rats [230]. These data suggest that eNOS in aorta from angiotensin-II infused rats is largely uncoupled due to BH_4_ deficiency and enzymatic function (e.g., NO synthesis as well as endothelium-dependent relaxation) is restored by supplementation with the BH_4_ precursor sepiapterin as well as PEG-SOD to prevent oxidative degradation of the sepiapterin-derived BH_4_ levels (but also autoxidation of sepiapterin in the solution). However, it should be noted that, in a model of experimental hypertension based on spontaneously hypertensive rats (SHR), this ex vivo treatment regimen was not successful and it failed to restore endothelial function (Figure 7) [230]. This observation would be compatible with previous results, indicating that endothelial dysfunction in old SHR is mainly based on vascular remodeling and not easily reversed by post-hypertensive therapy. Most studies on antihypertensive therapy in SHR rats start in young animals, in the pre-hypertensive state to prevent vascular remodeling, and in the development of hypertension and its adverse effects (reviewed in [23]). A number of reports suggests that therapeutic treatment with AT_1_-receptor-blockers, statins, and organic nitrates with pleiotropic antioxidant effects, improves the expression of GCH-1, and thereby normalizes BH_4_ levels, eNOS, and endothelial function in diabetes, hypertension, and nitrate tolerance [188,230,231,232]. 

However, despite strong evidence from preclinical and small-size cohort human studies on a therapeutic benefit of eNOS “recoupling” by administration of BH_4_ [233] or its precursors (e.g., sepiapterin or folate) [13], this concept was until now not translated to clinical therapy. Clinical evidence for a role of oxidative BH_4_ is based on the improvement of endothelial function in chronic smokers by BH_4_ but not by tetrahydroneopterin (NH_4_), which shares the same antioxidant properties with BH_4_ but is not a cofactor for eNOS (Figure 7) [233,234]. Likewise, supplementation with the BH_4_ analogue folic acid improves endothelial function in human subjects [235,236] or treatment with the BH_4_ precursor sepiapterin restored endothelial function in experimental hypertension as well as atherosclerosis [118,230]. Despite promising preclinical data on a protective role of BH_4_ in ischemic stroke models, a small cohort human study (multicenter randomized, double-blind, placebo-controlled trial) found no significant effects on the prognosis of 61 stroke patients by sapropterin (BH_4_ drug) therapy (200–400 mg twice per day for two months) [237], which may be related to the activation of iNOS (a potential pharmacological target in stroke) by increased BH_4_ levels. In a double-blind multicenter clinical trial 40 patients with cirrhosis and portal hypertension either received sapropterin (5 mg/kg/d for two weeks) or placebo, but hepatic blood flow, systemic hemodynamics, endothelial dysfunction markers, and liver function tests remained unchanged [238]. Other small cohort trials reported that a single oral dose of 10–20 mg/kg/d BH_4_ improves endothelial function in patients with cystic fibrosis or healthy subjects (58 subjects) [239], 12 patients with systemic sclerosis [240] and 33 patients with rheumatoid arthritis [241], just to mention some of them. Supra-nutritional doses of folate seem to prevent and help to control diabetic complications [242]. Therefore, the clinical data are so far inconclusive and warrant larger clinical trials on the cardio/cerebrovascular effects of BH_4_ analogs.

Also, the development of eNOS enhancers (by upregulation of eNOS mRNA and protein), despite highly promising preclinical data of AVE9488 on improvement of left ventricular remodeling in experimental MI [243] and AVE3085 on improvement of diastolic heart failure in rats [244], did so far not result in a new clinical therapy. eNOS enhancers would have yielded more beneficial data if the drugs not only up-regulate eNOS but also the BH_4_-synthesizing enzyme GCH-1. A reason for this might be that eNOS enhancement by upregulation is only successful when at the same time the BH_4_ levels are also increased (e.g., by up-regulation of GCH-1), otherwise the up-regulated eNOS will be uncoupled and produce superoxide instead of NO [245,246]. The eNOS enhancer AVE9488 enhanced the functional activity of bone marrow mononuclear cells upon ex vivo pretreatment, which could be exploited for cell therapy using “EPCs” [247] and prevented ischemic damage in mice [248]. In diabetic rats, eNOS enhancement by AVE3085 normalized altered hind limb blood flow and vascular inflammation [249]. AVE3085 also normalized endothelial function and blood pressure in spontaneously hypertensive rats [250], reduced oxidative stress and endothelial dysfunction in diabetic db/db mice [251], attenuated cardiac remodeling in mice [252], and prevented endothelial dysfunction by homocysteine in human vascular tissue [253]. Excessive platelet activation in rats with congestive heart failure was prevented by therapy with the eNOS enhancer AVE9488, most probably by improved platelet NO bioavailability [254]. The mechanism of eNOS enhancers via the upregulation of eNOS gene and protein are based on transcriptional activation via classical transcription factors but also on epigenetic pathways (see Section 3.4), mechanisms that are triggered by a number of natural compounds, such as polyphenols [255,256].

### 3.4. Epigenetic Regulation of eNOS Expression

In the vascular wall, eNOS is normally expressed in the endothelial cells but not in the smooth muscle cells (SMC). Epigenetic, chromatin-based mechanisms have been shown to be responsible for the cell type-specific expression of the eNOS gene [257]. The human eNOS promoter lacks a canonical TATA box and it does not contain a proximal CpG island [257]. A differentially methylated region (DMR) has been found in the native eNOS proximal promoter (−361/+3), with light methylation in eNOS-expressing (endothelial) cells and heavy methylation in non-expressing cell types (e.g., SMC) [258]. Consistently, the methyl-CpG-binding protein MeCP2, which is associated with transcriptional repression, is preferentially recruited to the eNOS proximal promoter in non-expressing cell types [259]. Consequently, transcription factors (Sp1, Sp3, and Ets1) and the transcriptional machinery (RNA polymerase II) are preferentially recruited to the eNOS proximal promoter in endothelial cells but not in SMC, despite the presence of these factors in SMC [258]. Treatment of non-endothelial cells with 5-azacytidine, a DNA methyltransferase (DNMT) inhibitor, leads to demethylation of the eNOS promoter and increased eNOS mRNA expression in these cells [258]. These results convincingly demonstrate the functional relevance of the eNOS proximal promoter DMR in controlling eNOS expression in a cell type-specific manner.

In addition, eNOS expression is also regulated by histone-based mechanisms. The eNOS proximal promoter in endothelial cells is enriched with H3K4 di- and trimethylation, as well as H3K9 and H4K12 acetylation, epigenetic marks are associated with transcriptional activation [259]. The low levels of histone acetylation in non-endothelial cells at the eNOS proximal promoter are consistent with the enrichment of histone deacetylase 1 (HDAC1) [259]. The functional relevance of these histone codes has been shown by the treatment of SMC with a pharmacological HDAC inhibitor. HDAC inhibition leads to enhanced acetylation of histones H3 and H4 at the eNOS proximal promoter and concomitantly increased eNOS mRNA expression in SMC [259,260]. These data demonstrate that epigenetic pathways (DNA methylation and histone modifications) are fundamental determinants of eNOS gene expression. Perturbations in these epigenetic pathways may lead to eNOS expression changes as contributing mechanisms in cardiovascular disease [257]. The pre- and postnatal periods are sensitive phases of developmental plasticity that are mediated by epigenetic modifications, whereas the sensitivity of the epigenome to the environment decreases during life as growth slows [261].

Adverse intrauterine conditions increase the risk of developing cardiovascular and metabolic diseases in adulthood. Intrauterine growth restriction (IUGR), for instance, is associated with endothelial dysfunction and cardiovascular risk in the offspring and the NO system has been implicated in these effects [262]. The expression levels of eNOS are changed in human endothelial cells that were isolated from umbilical arteries (HUAEC) and veins (HUVEC) from IUGR fetuses, with eNOS protein and mRNA levels increased in HUAEC, but decreased in HUVEC [262]. This is associated with a decrease in DNA methylation at CpG −352 at the eNOS promoter in IUGR-HUAEC, and an increase in IUGR-HUVEC. Silencing of DNMT1 expression normalizes eNOS expression in IUGR endothelial cells, indicating the causal role of DNA methylation in regulating eNOS expression under the condition of IUGR [262]. Recently, this finding has been extended to systemic fetal arteries [263]. IUGR in guinea pigs leads to the upregulation of eNOS in the aorta and femoral artery endothelial cells of the fetus, associated with a reduction of DNA methylation at a specific CpG (−170) of the eNOS promoter [263]. 

The increased eNOS expression in IUGR is likely to be a compensatory mechanism. This compensation, however, does not result in enhanced NO production. The reason for this futile compensation is attributable to a dysfunction of the eNOS enzyme. In the guinea pig IUGR, the upregulated eNOS has been shown to be associated with a reduced eNOS phosphorylation at serine 1177 [263]. In a low-protein diet-induced IUGR rat model, the upregulated eNOS in the aorta of the offspring is found in an uncoupled state because of a deficiency of L-arginine [264]. Uncoupled eNOS produces superoxide instead of NO [264,265], which may represent the eNOS-mediated mechanism contributing to the endothelial dysfunction that was observed in these IUGR models. On the other hand, the perinatal period may also represent an opportunity for the cells to be reprogrammed to a ‘normal type’ by interfering the epigenetic machinery. Treatment of IUGR guinea pigs with the glutathione precursor N-acetylcysteine leads to a normalization of both the DNA methylation at eNOS promoter and eNOS mRNA levels in fetal artery endothelial cells, which is associated with a restoration of eNOS-dependent relaxation in aorta and umbilical arteries [263]. Our group has shown in a recent study that the treatment of spontaneous hypertensive rats (a rat model of genetic hypertension) during the pregnancy and lactation periods with PETN leads to an improvement of endothelial function and a persistent blood pressure reduction in the female offspring. These programming effects are likely to be attributable to long-lasting gene expression changes (including eNOS) resulting from epigenetic mechanisms [266].

## 4. Conclusions

Endothelial (vascular) dysfunction is a hall-mark of most cardiovascular risk factors and diseases. Besides the classical risk factors and diseases (e.g., hypertension, hyperlipidemia, diabetes, smoking, alcohol abuse) that are partly based on genetic predisposition but also behavioral (life-style) causes, there are also novel risk factors in the physical environment (e.g., traffic noise and air pollution exposure), all of which are associated with the development of endothelial dysfunction as well as the pathomechanisms that are highlighted in Figure 8. Oxidative stress and inflammation represent common key pathways associated with these risk factors and diseases that trigger the described down-stream pathomechanisms, namely adverse redox regulation of eNOS function (potentially not only in endothelial cells but also in PVAT) and related pathways. Targeting vascular (endothelial) dysfunction still represents an attractive pharmacological approach due to its central involvement in the progression of most cardio- and cerebrovascular diseases as well as inflammatory processes. In addition to the classical cardiovascular drugs that have beneficial effects on vascular (endothelial) function, such as ACE inhibitors and statins, new therapeutic strategies that are based on modulation of the vascular the direct targeting of the ^•^NO/cGMP signaling cascade are attractive. Despite of the failure to use chronic pharmacological ^•^NO replacement therapy in the clinics, modulation of eNOS function itself still represents a promising pharmacological strategy, especially due to the substantial redox-regulation of eNOS activity. The clinical value of BH_4_ supplementation is insufficiently studied in men so far and more large-scale studies are needed. The other “redox switches” in eNOS may be beneficially modulated by (site-specific) antioxidant therapy but this remains to be demonstrated. As shown for oxidatively inactivated sGC, a highly specific eNOS activator or stimulator (not based on upregulation at the genetic or protein level) may fulfill a similar role for uncoupled/dysfunctional eNOS and only activate the damaged, oxidatively inactivated form of eNOS, which would represent a safe way to treat endothelial dysfunction without the risk of overdosing. Also, the potential of eNOS enhancers requires further exploration in clinical studies, maybe in combination with sapropterin and/or antioxidant therapy in order to prevent the uncoupling of overexpressed eNOS protein. The role of eNOS function in PVAT needs further exploration in order to include PVAT eNOS functions into future pharmacological development of eNOS activators, stimulators, and enhancers. Finally, epigenetic therapy shows beneficial effects in animal models but it requires more extensive investigation to transfer this to a clinical setting.

## Figures and Tables

**Figure 1 ijms-20-00187-f001:**
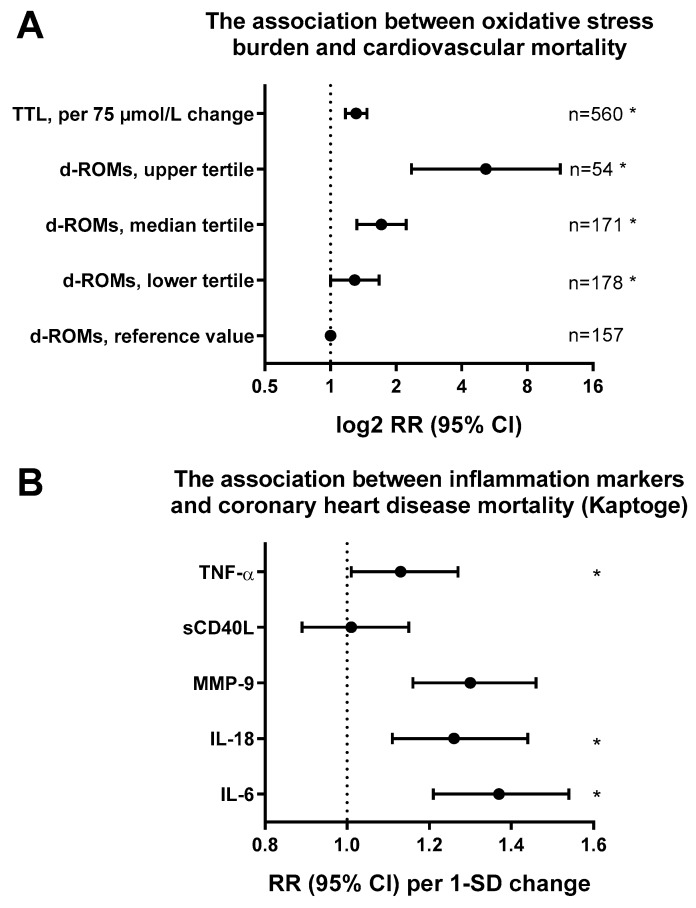
Impact of oxidative stress and inflammation markers on cardiovascular events or mortality. (**A**) Associations of derivatives of reactive oxygen metabolites (d-ROMs) levels and total thiol levels (TTL) with cardiovascular disease-specific mortality Adjustments for other confounders as described for model 1 (age and sex). d-ROMs groups [Carratelli Units]: reference, <340; T1, 341–400; T2, 401–500; T3, >500. *, *p* < 0.05 versus reference value. Graph was generated from tabular data in Schöttker et al. *BMC Med.* 2015 [38]. (**B**) Hazard ratios for all coronary heart disease in correlation with markers of inflammation (cytokines and chemokines: IL-6, IL-18, MMP-9, sCD40L, and TNF-α) in the Danish Research Centre for Prevention and Health cohort (1514 subjects, 833 cases). Adjustment for sex and age, log-transformed baseline levels of cytokines. Redrawn from tabular data in Kaptoge et al. *Eur. Heart J.* 2014 [46].

**Figure 2 ijms-20-00187-f002:**
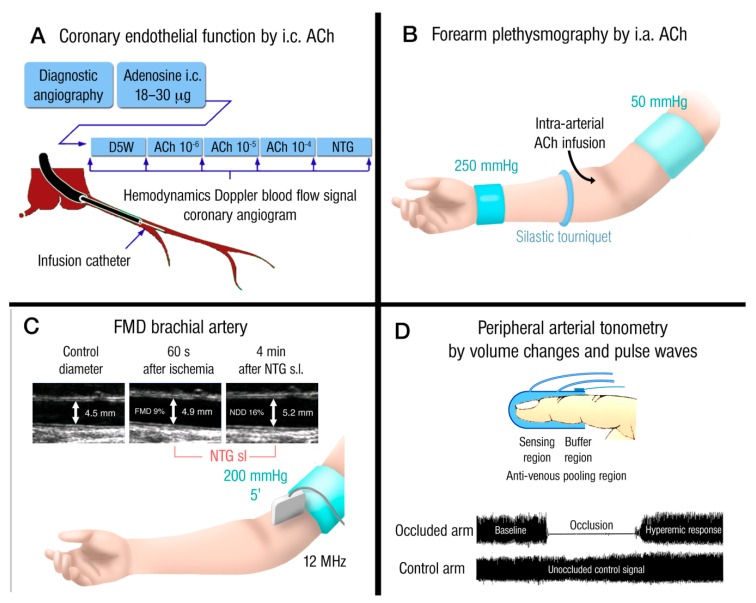
Invasive and non-invasive methods for the determination of endothelial function. (**A**) Acetylcholine (ACh)-dependent vasoreactivity of coronary vessels caused by intra-coronary ACh infusion. Vasodilation and stenotic areas are monitored by angiographic imaging and Doppler ultrasound (blood flow). (**B**) ACh-dependent vasoreactivity of capacity vessels of the forearm upon intra-arterial ACh infusion. Vasodilation is recorded by Doppler ultrasound (diameter and blood flow). (**C**) Flow-mediated dilation (FMD) of capacity vessels of the forearm (brachial artery) upon occlusion/ischemia and reperfusion/hyperemia. Vasodilation is recorded by Doppler ultrasound (diameter and blood flow). Maximal vasoreactivity/dilation is determined by sublingual administration of nitroglycerin (NTG). Adapted from [86]. With permission of Taylor & Francis (Abingdon, Oxfordshire, UK). Copyright © 2008, Rights Managed by Taylor & Francis (Abingdon, Oxfordshire, UK). (**D**) Peripheral arterial tonometry (PAT) measures volume changes and pulse waves by a finger probe that assesses digital volume changes and pulse waves that can be detected after induction of reactive hyperemia. Adopted from Daiber et al., *Br. J. Pharmacol.* 2017 [13]. With permission of the publisher. Copyright © 2016, John Wiley and Sons (Hoboken, NJ, USA).

**Figure 3 ijms-20-00187-f003:**
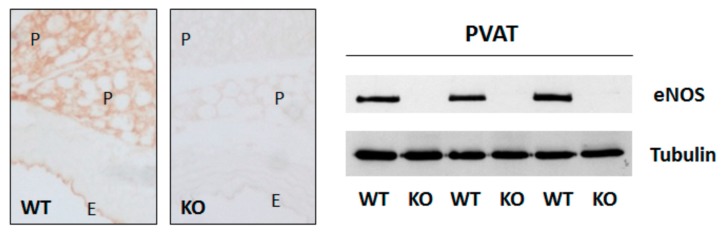
Expression of endothelial nitric oxide synthase (eNOS) in perivascular adipose tissue (PVAT) adipocytes. eNOS immunohistochemistry staining and western blot analyses were performed using PVAT-containing aorta samples from C57BL/6J wild-type mice (WT) or global eNOS knockout mice (KO). E and P indicate endothelium and PVAT respectively. Reproduced from Xia et al. *Br. J. Pharmacol.* 2017 [107], an open access article under the terms of the Creative Commons Attribution-Non-Commercial License CC BY-NC. Copyright © 2017, handled by John Wiley and Sons (Hoboken, NJ, USA).

**Figure 4 ijms-20-00187-f004:**
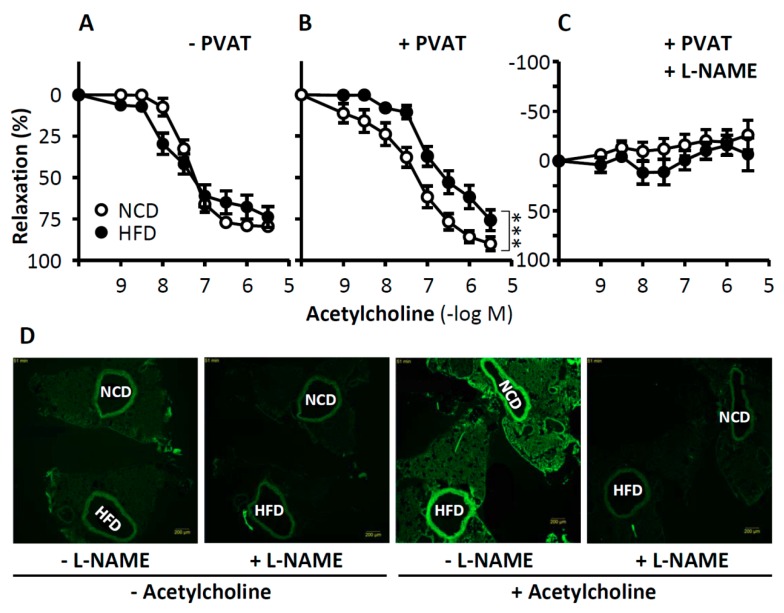
Role of PVAT in obesity-induced vascular dysfunction. C57BL/6Jmice were fed a high-fat diet (HFD) or normal control diet (NCD) for 20 weeks starting at the age of eight weeks. The vasodilator response to acetylcholine (**A**–**C**) was performed in noradrenaline-precontracted aorta with or without PVAT in the absence or presence of the NO synthase inhibitor L-NAME. *** *p* < 0.001, *n* = 8. To detect PVAT NO production, NCD and HFD aorta samples were mounted back-to-back on the same slide to guarantee identical staining conditions for the two samples (**D**). NO production in PVAT-containing aorta was determined by 4,5-diaminofluorescein diacetate (DAF-2 DA) staining. Magnification bar is equal to 200µm. From Xia et al. *Arterioscler. Thromb. Vasc. Biol.* 2016 [98]. With permission of Wolters Kluwer Health, Inc. (Philadelphia, PA, USA). Copyright © 2015, Wolters Kluwer Health (Philadelphia, PA, USA).

**Figure 5 ijms-20-00187-f005:**
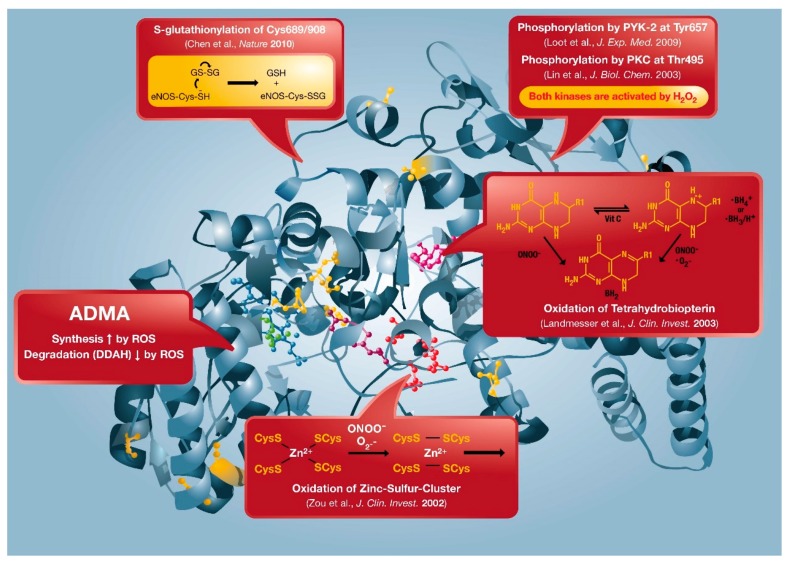
Redox switches in eNOS. X-ray structure of human eNOS based on the protein database entry **3NOS** (DOI:10.2210/pdb3nos/pdb) using the PyMOL Molecular Graphics System Version 1.2r1 (DeLano Scientific LLC). The boxes represent the “redox switches” in eNOS such as S-glutathionylation, PKC- and PYK-2 dependent phosphorylation, oxidative BH_4_ depletion, disruption of the zinc-sulfur-cluster, as well as ADMA synthesis/degradation, all of which contribute to the regulation of its enzymatic activity. GSH, glutathione; GSSG, glutathione disulfide. From Schulz et al., *Antioxid. Redox Signal.* 2014 [20]. With permission of Mary Ann Liebert Inc. (New Rochelle, NY, USA). Copyright © 2014, Mary Ann Liebert, Inc. (New Rochelle, NY, USA).

**Figure 6 ijms-20-00187-f006:**
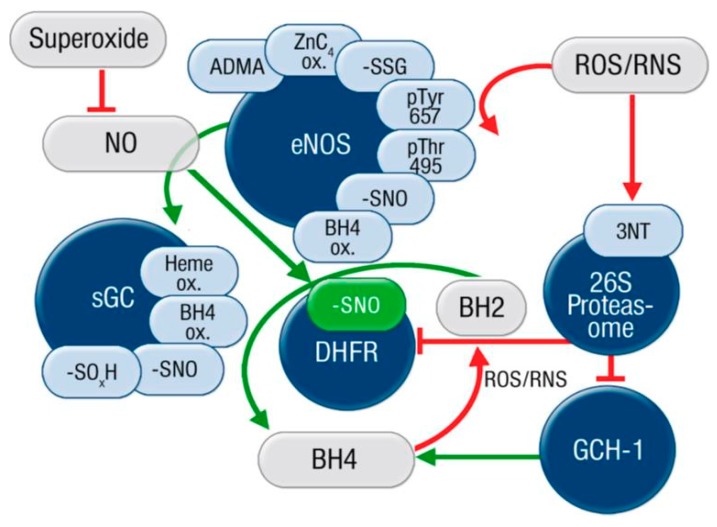
Summary of redox regulatory pathways of vascular tone. Endothelial nitric oxide synthase (eNOS) and soluble guanylyl cyclase (sGC) are inactivated by a number of redox switches. Reactive oxygen and nitrogen species (ROS/RNS) also activate the 26S proteasome via 3-nitrotyrosine (3NT) modification leading to degradation of the tetrahydrobiopterin (BH_4_) synthase GTP-cyclohydrolase (GCH-1) and the BH_2_ to BH_4_ recycling enzyme dihydrofolate reductase (DHFR). BH_4_ is an essential cofactor of eNOS and also prevents oxidative inactivation of the sGC. NO is inactivated by superoxide and eNOS-derived NO prevents proteasomal DHFR degradation upon tyrosine nitration of the 26S proteasome. From Münzel and Daiber, *Arterioscler. Thromb. Vasc. Biol.* 2015 [135]. With permission of Wolters Kluwer Health, Inc. (Philadelphia, PA, USA). Copyright © 2015, Wolters Kluwer Health (Philadelphia, PA, USA).

**Figure 7 ijms-20-00187-f007:**
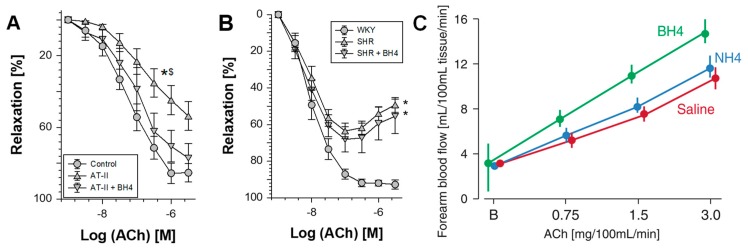
Improvement of endothelial function ex vivo by sepiapterin and in vivo by BH_4_. The effects of sepiapterin (100 µmol/L), a BH_4_ precursor, and polyethylene glycolated-superoxide dismutase (100 U/mL) pretreatment of aortic rings from angiotensin-II (AT-II)–infused rats (**A**) or spontaneously hypertensive rats (SHR, **B**) for 1 h were determined in separate experiments. Data shown are representative for at least three animals or at least six aortic rings/group. *p* < 0.05: * vs. WKY/Control; ^$^ vs. AT-II/BH_4_. The statistics were based on one-way-ANOVA comparison of pD_2_-values and efficacies but also on comparisons of all concentrations in all groups by two-way-ANOVA analysis (for sake of clarity significance is not shown for all data points). From Schuhmacher et al., *Hypertension* 2010 [230]. With permission of Wolters Kluwer Health, Inc. (Philadelphia, PA, USA). Copyright © 2010, Wolters Kluwer Health (Philadelphia, PA, USA). (**C**) Effect of tetrahydrobiopterin (BH_4_) and tetrahydroneopterin (NH_4_) on the acetylcholine (ACh) dose-response relationship in chronic smokers. BH_4_ significantly improved ACh dose-response relationship, whereas NH_4_ was ineffective. Modified from Heitzer et al., *Circ. Res.* 2000 [233] and published in Schulz et al. *Antioxid. Redox Signal.* 2008 [119]. With permission of Mary Ann Liebert, Inc. (New Rochelle, NY, USA). Copyright © 2008, Mary Ann Liebert, Inc. (New Rochelle, NY, USA).

**Figure 8 ijms-20-00187-f008:**
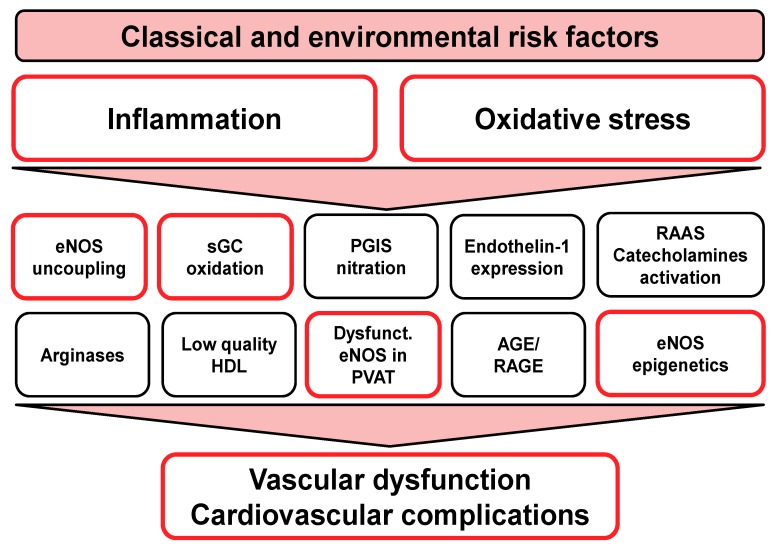
Initiators of endothelial (vascular) dysfunction. Inflammation and oxidative stress are strong triggers of endothelial (vascular) dysfunction. Of note, pathomechanisms of classical and environmental risk factors (see top of the scheme) converge to a certain extent at the level of inflammation and oxidative stress to further trigger the down-stream dysregulation and damage: eNOS uncoupling, sGC oxidation (maybe also imbalanced phosphodiesterase expression/activity), prostacyclin synthase (PGIS) nitration and inactivation, redox-triggered endothelin-1 (ET-1) signaling, activation of the renin-angiotensin-aldosterone system (RAAS) and sympathetic activation (catecholamines), and finally, AGE/RAGE signaling. In addition, dysregulated arginase metabolism decreases levels of the eNOS substrate L-arginine. The quality of high density lipoprotein (HDL) changes under oxidative stress conditions and metabolic disease. Epigenetic mechanisms also contribute to the (dys)regulation of eNOS expression and activity. The boxes with red lining are discussed in detail in the present review. Modified from Daiber et al., *Br. J. Pharmacol.* 2017 [13]. With permission of John Wiley and Sons. (Hoboken, NJ, USA). Copyright © 2016, John Wiley and Sons (Hoboken, NJ, USA).

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
