# Peer review of "New Therapeutic Implications of Endothelial Nitric Oxide Synthase (eNOS) Function/Dysfunction in Cardiovascular Disease"

_ijms, 2019, doi:10.3390/ijms20010187_

Round 1

Reviewer 1 Report

This is a helpful review on Redox/eNOS and related cardiovascular diseases. I would like to suggest the following:

Major comments:

Fig. 2. This figure is too busy. Increase font size. Reduce words in the diagram.

Fig. 5 blue, green, purple seem to be the key. But blue & green are hidden in the woods so readers without magnifier wouldn't be able to play a hide & seek game here. Use of different colors does not help. Tone down the background color so that the foreground (your focus) can be more visible while your background helps what you aimed to show.

Fig. 6 (clear! Change Fig. 5 like this one)

Fig. 7 A & B - high resolution & sharp   C - low resolution and less clear graphs...why?

Fig. 8 Simplify words on the top box (currently, too busy: you lose your focus by throwing away everything. Be succinct. Please summarize your take home message so that we remember.

Abstract: background (first 2 sentences) too long. e.g. GBD story. Please be concise. Rather, expand more on what you really reviewed in this manuscript at the end (1-2 sentences).

Structure: 1. CVD 2. Redox (eNOS) 3. Future therapeutics (eNOS) - the title of each does not connect well each other. What can be a more appropriate title for section 1? Be specific with your expertise. Cardiovascular disease is too broad to be used as the section 1.

Minor suggestions:

1.1 Authors assumed that readers already knew GBD. However, that is a jump. Who cares GBD? Briefly explain why you began your story with GBD papers. Would GBD be an accurate or important or valuable study for you to mention in the beginning (all of a sudden)? Why were you so motivated about GBD?

Please be kind enough to explain what "communicable or non-communicable disease" are. The one who read the GBD study would know what those are but others who did not would not. I know you read such papers (GBD) but why readers should care unless you mentioned.

pp.2 line 35 define NADPH (is it important? is it enzyme? why did you introduce?)

pp.3 line 43 define Nox2

pp.3 line 48 define RAG-1-/- (who cares about this mouse? Nobody cares unless you explain)

pp.6 Be consistent. Use of both Noradrenalin and norepinephrine confusing.

pp. 8 line 1-10 larger font size; while line 11-34 smaller font size (be consistent)

Thank you.

Author Response

Our Point-by-Point Response can be found in the attached pdf-file

Reviewer 2 Report

In the present study, Daiber et al. comprehensively summarized and discussed the physiological and pathophysiological role of eNOS in cardiovascular field. The review is balanced, comprehensive and very well written. Several minor comments need to be addressed.  

1.     The title is not very on focus. All the discussion the authors made is in cardiovascular field. In addition, the eNOS dysfunction is rather confusing, should necessarily be “eNOS uncoupling” compared to “dysfunction”. Would not be it better to rephrase it to something like “new therapeutic implications of uncoupled eNOS in cardiovascular field”? The same questions for “eNOS dysfunction” throughout the entire manuscript. The full name for eNOS in the title may also be considered?

2.     The sentence is too long on page 3, line 14-19. Please rephrase it. The same for the sentence on page 3, line 35 to 37.

3.     The authors mentioned vascular cells that generate ROS accounting for endothelial dysfunction at section 1.3 on page 3. Recent interesting studies showed red blood cells serve as ROS pool for cardiovascular dysfunction in diabetes (Yang J, JACC Basic Transl Sci 2018 PMID:30175269; Zhou J, J Am Coll Cardiol 2018 PMID:30092954). Should the authors consider this aspect in the related section.

4.     On page 5, line 25. Can authors add “and experimentally” after “clinically”, since the figure 2F was done in experimental animals?

5.     The authors discussed the measurement of endothelial function in human that is evaluated by Acetylcholine only. As serotonin is also commonly used for this purpose in human, additional examples with serotonin studies need to be discussed or image shown in the figures.

6.     On page 6, line 22, Figure 4 should be Figure 4D. On page 7, line 6, Figure 4 should be Figure 4A-C). Also please enlarge or remove the white legends in the background of Figure 4D.

7.     In figure 5, the brown color cannot be visualized (see the legend “zinc ion (brown)”. In figure 6, ROS/RNS also activate the 26S proteasome, I guess the RED arrow should go closer to the blue “26S proteasome” in the figure. On page 10 line 19, please update the “???”.

8.     Despite the authors did mention the arginase throughout the manuscript e.g. on page 10 line 27 and the Figure 8, the arginase and eNOS relation should be a bit more discussed in the text in related section (maybe in the section 2.5?).

9.     Typo error? On page 15 line 14 for “vasculo”.

10.  The abbreviations need to be presented as their full names at the first time. For example, HUVEC on page 16 line 41. Double check the entire manuscript please.

Author Response

(The authors gave the same response as above.)

Round 2

Reviewer 1 Report

I have no objection any more. Check spelling again and go ahead for publication. Thank you.

Author Response

We did a final spell checking of the manuscript and included last minor language corrections.